# Grain Yield Performance and Quality Characteristics of Waxy and Non-Waxy Winter Wheat Cultivars under High and Low-Input Farming Systems

**DOI:** 10.3390/plants11070882

**Published:** 2022-03-25

**Authors:** Jurgita Cesevičienė, Andrii Gorash, Žilvinas Liatukas, Rita Armonienė, Vytautas Ruzgas, Gražina Statkevičiūtė, Kristina Jaškūnė, Gintaras Brazauskas

**Affiliations:** Institute of Agriculture, Lithuanian Research Centre for Agriculture and Forestry, 58344 Kedainiai, Lithuania; andrii.gorash@lammc.lt (A.G.); zilvinas.liatukas@lammc.lt (Ž.L.); rita.armoniene@lammc.lt (R.A.); vytautas.ruzgas@lammc.lt (V.R.); grazina.statkeviciute@lammc.lt (G.S.); kristina.jaskune@lammc.lt (K.J.); gintaras.brazauskas@lammc.lt (G.B.)

**Keywords:** *Triticum aestivum* L., waxy wheat, cultivar, low-input farming, intensive farming, yield, grain quality, flour—dough rheology, starch properties, RVA

## Abstract

Waxy starch with a modified amylose-to-amylopectin ratio is desired for a range of applications in food and non-food industries; however, yield performance and grain quality characteristics of waxy wheat cultivars are usually inferior in comparison to advanced non-waxy cultivars. In this study, we compared waxy (‘Eldija’, ‘Sarta’) and non-waxy (‘Skagen’, ‘Suleva DS’) winter wheat cultivars grown under high and low-input farming systems over two cropping seasons by evaluating their yield and grain quality, including flour, dough, and starch physicochemical properties. The yield of waxy cv. ‘Sarta’ was significantly lower compared to the non-waxy cultivars across all trials; however, waxy cv. ‘Eldija’ had a similar yield as non-waxy cultivars (except under high-input conditions cv. ‘Skagen’). Moreover, no significant differences were observed between protein and gluten content of waxy and non-waxy cultivars. Low amylose content typical for waxy wheat cultivars highly correlated (r ≥ 0.8) with lower falling number, flour yield and sedimentation values, lower nitrogen % used for grain, higher flour water absorption and flour particle size index. In general, properties dependent on starch structure demonstrated consistent and significant differences between both starch types. The prevailing heat waves during the grain filling period decreased grain test weight but increased protein and gluten content and caused gluten to be weaker. Dough development time at these conditions became longer, dough softening lowered and starch content decreased, but A-starch, starch peak and final viscosity values increased. Low-input farming had a negative effect on grain yield, grain nitrogen uptake and grain test weight but increased phosphorus content in grain. The unique dough mixing properties of waxy cultivar ‘Eldija’ suggest that it could be used in mixtures along with non-waxy wheat for dough quality improvement.

## 1. Introduction

Wheat is one of the staple crops grown worldwide along with maize and rice. About 35% of the total world population regularly consumes wheat-based food. Starch is the main component, existing in the endosperm of wheat grain and comprises about 60–70% of the whole grain and 65–75% of white flour. Wheat starch consists of two glucose polymers: linear amylose and branched amylopectin with the ratio range of 25–28 and 72–75%, respectively [1]; however, advances in biotechnology in recent years have enabled increasing the ratio up to 80% for amylose or 99–100% for amylopectin of some mutant genotypes of wheat [2,3,4]. The amylose/amylopectin ratio is the main factor affecting physiochemical properties of starch and the quality of end-use products. The amylose of wheat endosperm is encoded primarily by granule-bound starch synthase (GBSS1) enzyme at three loci on chromosomes 7AS (*Wx-A1*), 4AL (*Wx-B1*) and 7DS (*Wx-D1*). The presence of null/non-functional alleles at all three loci confer a fully waxy type of endosperm starch (<1% of amylose), whereas the null alleles in one or two loci provide a partial waxy type [5,6,7].

The development of waxy wheat cultivars with a modified amylose-to-amylopectin ratio of starch has provided a novel raw material and opened new opportunities for food and non-food industries [5,8]. Unique physicochemical properties of waxy starch prolong the shelf life of baked goods and improve the quality of frozen dough [1,9,10] because even after the freezing cycle, waxy wheat flour maintains higher stickiness and extensibility than non-waxy flour [11,12]. The lower gelatinization temperature of waxy starch raises its potential market demand as a thickener for microwaved foods, saving energy due to the reduced cooking time [13,14]. Moreover, partial waxy wheat flour increases the elasticity and stickiness of the dough texture for noodle production [15,16]. Udon noodles are one of the most popular Japanese staple foods, which have become popular and consumed worldwide. Considering the high market demand, two separate milt classes for producing noodles have been defined in Australia [17]. Moreover, the waxy starch has several non-food uses, as a component of glues and adhesives, textiles, pharmaceuticals, or paper to improve or add new properties [5]. Additionally, the waxy starch is more efficient for ethanol production in comparison to the non-waxy wheat starches [18].

Due to an increase in the consumption of food, feed, fuel and to meet global food security needs for the rapidly growing human population, extensive breeding and intensification of agricultural practices for the past half-century have been applied. However, the intensive farming systems with the overuse of fertilizers and pesticides have negatively impacted ecosystems and provoke the expansion of agricultural land used for environmentally-friendly farming systems. There is a growing consensus that organic farming and the principles of integrated agricultural systems are crucial for mitigating climate changes and attaining sustainable agriculture [19,20]. The regulations concerning pesticide management are different; but in general, slight tendencies in decreased pesticide use have been observed around the world [21].

Graybosch et al. [22] determined that flour yield, falling number and starch properties were negatively affected by waxy starch, and it did not depend on genetic background and environments. However, it was found that waxy mutation does not provide a significant effect on such agronomic traits as kernel weight and grain yield [22,23]. Sharma et al. [24] and Vignaux et al. [25] found no association of waxy genes with protein content in durum wheat. According to the study of Graybosch et al. [10], gluten index scores of 22 waxy lines (44%) did not differ significantly from the highest ranking non-waxy check.

Based on previous studies, it can be hypothesized that waxy cultivars with superior yield performance and protein quality properties could be developed. While flour yield and falling number might be difficult to improve due to strong association with waxy genes, other traits such as grain yield and protein-related quality attributes are more amenable for improvement.

The objective of this study was to compare the yield and grain quality, including flour, dough and starch physicochemical properties, across genotypes, environments and farming systems by studying two waxy and two non-waxy winter wheat cultivars under low-input and intensive farming systems.

## 2. Materials and Methods 

### 2.1. Plant Materials and Experimental Design

Four winter wheat (*Triticum aestivum* L.) cultivars adapted to the northern European region were used in the study: two non-waxy—‘Skagen’ (registered in 2011, developed in DK), ‘Suleva DS’ (2019, LT) and two waxy (wx)—‘Eldija’ (2021, LT) and ‘Sarta’ (2021, LT). Non-waxy cultivars were used as control cultivars.

Field trials were conducted during 2017–2019 at the Lithuanian Research Centre for Agriculture and Forestry (LAMMC) in Akademija, Kedainiai district, Lithuania (55°39′ N 23°57′ E) under low-input (L) and intensive (I) or high-input farming systems. The soil was light loam Endocalcari-Epihypogleyic Cambisol. The topsoil (0–30 cm) pH was low acid/neutral (5.7/7.0 in L/I trials), close to moderate in humus (18 g kg^−1^), high in available phosphorus (210 ± 7 mg kg^−1^ P_2_O_5_) and from high to moderate in available potassium (213/130 mg kg^−1^ K_2_O in L/I trials). The field experiment was designed in three replications (plot size 11 × 1.6 m), each farming system was grown in a separate block, where replications of field plots were randomized. Winter wheat was sown at the seed rate of 4.5 million ha−1 in the end (27–28th, 2017) or middle (10th/17th, 2018 in L/I) of September after the black fallow. In every year, complex mineral fertilizers (N15P50K100) were applied in the whole experimental field before sowing. Nitrogen fertilizers (ammonium nitrate) were applied after resumption of spring vegetation (at 10 April 2018; 25 March 2019) and when plants reached stem elongation stage (at 10 and 2 May, in 2018 and 2019, respectively). The rates of N100 + 30 and N100 + 100 were used in low-input (L) and intensive farming (I) systems, respectively. Weeds were controlled by the recommended herbicides in the autumn and spring in both growing systems. Seed treatment, plant growth regulators, fungicides and insecticides were applied only at the recommended rates and time in the intensive farming system.

Heading time varied among years, farming systems and cultivars. In 2018, plants reached the heading stage on 22–31 May (L trial) and 26 May–1 June (I trial). In 2019, plant heading was recorded on 30 May–5 June (L trial) and 26 May–2 June (I trial). The cultivars differed by growth rate. Wx ‘Eldija’ and wx ‘Sarta’ plants reached the heading stage 5–9 and 4–8 days earlier, and ‘Suleva DS’–in 2–4 days earlier than medium late standard cultivar ‘Skagen’. Harvesting was carried out when the majority of plants had reached full maturity on 20 and 23 July in the L trials (2018 and 2019, respectively) and 3 days later in the I trials. 

### 2.2. Weather Conditions

The elevation of the experimental area was 82 m above sea level, belonging to the mid-latitude climate zone in the southwestern subregion of the Atlantic continental forest area. According to data from the Dotnuva Meteorological Station (55°23′49.0″ N 23°51′55.0″ E), the climatic conditions are characterized by the long-term (1924–2019) annual temperature (year average is 6.5 °C) and precipitation (year average is 570 mm). Both experiment years were warmer, exceeding the long-term average by 2.3 °C and variable in terms of precipitation (Figure 1). The autumn of 2017 was extremely wet, while the autumn of 2018 was dry. The winters were quite mild; the spring meteorological conditions were variable—dry and windy weather prevailed. The spring of 2019 was later and cooler compared to 2018. The average air temperature in April was 3.9 °C higher than long term average in 2018 and 2.9 °C in 2019, while May of 2019 was even warmer by 4.5 °C compared to the long term average. Plants experienced heat waves during the grain filling period, with maximum temperatures rising above 25 °C after 10 July in 2018 and especially in June of 2019. Both experimental years were characterized as dry; drought period was recorded in May and July of 2018 and April and June of 2019. 

### 2.3. Determination of Wheat Grain Yield and Quality Characteristics

#### 2.3.1. Grain Yield Characteristics

The grain yield (GY) of wheat was estimated at standard moisture (14%). N use efficiency (NUE) was obtained by calculating kg grain production per N kg of fertilizer applied (means how much 1 kg of N in fertilizers gave kg of grain). Grain N yield (kg per ha in dry matter—DM) and N percent of fertilizers used for grain N yield production were also calculated.

#### 2.3.2. Grain Quality Characteristics and Chemical Composition 

Grain density was measured using the test weight method and was expressed as the weight of the grain in a specified volume (ISO 7971-2). Particle size index (PSI) for wheat grain hardness (AACC 55-30) was analyzed using laboratory mill LM3303 (Perten Instruments, Hägersten, Sweden) for flour preparation and 0.075 mm sieve for its sifting with Promylograph S1 (Max egger, Blasen, Austria). The content of nitrogen (N) and phosphorus (P) was evaluated in the sulphuric acid digestates. Samples for total N determination were analyzed using the Kjeldahl method with a Kjeltec 1002 system (Tecator AB, Hoganas, Sweden). N content calculation for a crude protein was based on multiplication of the N result by the conventional factor 5.7 (ISO 20483). The content of P was quantified spectrophotometrically by a colored reaction with ammonium molybdate–vanadate at a wavelength of 430 nm on a spectrophotometer Cary 50 UV-Vis (Varian Inc., Palo Alto, CA, USA). The flour sedimentation index was determined by the Zeleny method, in accordance with ISO 5529; flour for sedimentation was prepared using the Sedimat (Brabender, Duisburg, Germany) laboratory mill. The falling number (FN) was analyzed using falling number 1500 equipment (Perten Instruments, Hägersten, Sweden) by Hagberg method described in ICC 107/1. The wet gluten content was determined by hand washing method (LST 1522). The gluten dry content and quality, assessing the gluten index—GI, were analyzed using the Glutomatic System (Perten Instruments, Hägersten, Sweden) in accordance with the standardized Perten method (ICC 155); wet gluten absorption was calculated from wet and dry gluten content (when the wet gluten is equated to 100%). 

Sample preparation for a whole meal analysis (N, P, falling number, gluten parameters determination and for starch isolation) was carried out by grinding the wheat grains using LM3100 (Perten Instruments, Hägersten, Sweden) mill. The data on the chemical composition of the grains (N, protein, P, starch content) were recalculated on DM basis. Gluten content was adjusted to an 86% DM basis.

#### 2.3.3. Flour and Dough Quality Characteristics

The white flour was prepared with a Quadrumat Junior mill using a 70GG sieve (Brabender, Duisburg, Germany) after adjusting grain moisture content to a 14.5% moisture basis. Flour yield (extraction) was estimated. Rheological properties of flours were determined by measuring the resistance of the dough against the mixing. Water absorption, development time, stability, degree of softening at 10 min after mixing started (DS10), degree of softening at 12 min after peak (DS12) and farinograph quality number were measured by a Brabender’s farinograph with a mixer for 50 g of flour, using slow blade rotation speed (63 min^−1^) and measurement control system software 2.5.17 (Brabender, Duisburg, Germany); analysis duration time was 20 min. Analyses were performed in accordance with ICC 115/1 and ISO 5530-1.

#### 2.3.4. Starch Quality Characteristics

Starch isolation was performed with a modified version of the dough ball washing method [26], using a Glutomatic System (Perten instruments, Hägersten, Sweden) for starch/gluten separation with a whole meal washing program. After washing, the starch slurry was passed thought a 75 μm nylon sieve, centrifuged with Rotofix 32A (Xettich, Tuttlingen, Germany) at 3600 g for 10 min and the obtained supernatant discarded. Starch tailings (creamy color) in the upper layer of the pellet were carefully removed. The cleaning of the tailings was repeated using distilled water, stirring and centrifugation. The obtained starch pellets were dried with air convection at 40 °C and crushed using a vibratory micro mill Pulverisette 0 (Fritsch, Idar-Oberstein, Germany) before further analysis.

A-type starch granules (percentage volume, when diameter >10 μm) were measured by the particle size distribution in wet starch suspensions using the Hydro 2000MU module with laser scattering Mastersizer 2000 instrument equipped with Malvern application software version 3.20 (Malvern Instruments, Worcestershire, UK). The particles were assumed to have a refractive index of 1.52; distilled water was used as dispersant. 

The amylose content of isolated starch was analyzed by using the iodine-binding spectrophotometry-based method according to Zhu et al. [27] with slight modifications: 1 mL ethanol was added to 0.1 g starch sample, mixed and left approximately for 1 h. After suspension, the mixture was supplemented with 9 mL of 1 mol L^−1^ NaOH, mixed thoroughly by vortex and left in 35 °C incubator with easy stirring overnight. The dispersed sample (smooth and free of lumps) then was transferred to a 100 mL volumetric flask and diluted. One milliliter of well-mixed sample was transferred to 50 mL volumetric flask and filled with 25 mL of distilled water; 0.5 mL of 1 mol L^−1^ acetic acid was added, followed by 1.0 mL of 0.2% iodine reagent. The solution was diluted to 50 mL and the absorbance was measured after 20 min at 620 nm wavelength on a spectrophotometer Cary 50 UV-Vis (Varian Inc., Palo Alto, CA, USA). The determination of amylose content was calculated according to a standard curve developed by similar analysis using different ratios of amylose and amylopectin blends. High amylose (66%, from amylose/amylopectin assay kit K-AMYL, Megazyme, Bray, Ireland) and amylopectin (0% amylose, ChromaDex, Lot No. 00001665-00, Los Angeles, CL, USA) starch were used as standards, respectively.

The pasting properties of winter wheat starch were measured using the rapid visco-analyzer (RVA) (Tech Master, Newport Scientific, Warriewood, Australia) controlled with Thermocline software program. The analysis was performed with 13 min standard RVA profile (STD1), using 160 rpm rotor speed and programmed heating—cooling cycle (50 °C–95 °C–50 °C) in accordance with ICC 162. Parameters, including the viscosity peak, time and pasting temperature (at rise in viscosity), trough (minimum viscosity at 95 °C), breakdown (difference of peak and trough viscosity), final viscosity (viscosity at 50 °C) and setback (difference of final and trough viscosity) were recorded.

All whole-kernel, flour, dough and starch quality characteristics in each sample were determined in 2–3 replicates. Analyses were performed at the chemical research laboratory of the Institute of Agriculture, LAMMC. 

### 2.4. Statistical Analysis 

Collected data were subjected to a three-way analysis of variance (ANOVA). The procedure was performed considering the factors—wheat type (or cultivar), year and farming system—as fixed factors. Significant differences between factors were determined by F-test at *p* < 0.05, *p* < 0.01 probability levels. Standard error of the mean (*SE*) was used to represent error values and error bars. Significantly different means were calculated by Tukey’s studentized range test at *p* < 0.05, where means with the same letter are not significantly different.

Principal component analysis (PCA) was applied for reduction of the complexity of data sets to a small number of independent principal components, for assessment of the association between groups of variables, and an understanding of the primary components, which contribute to the underlying variability of data set.

Statistical analyses were performed with Statistica software, version 7.1 (StatSoft Inc., Tulsa, OK, USA) and SAS, version 7.1 (SAS Institute, Cary, NC, USA).

## 3. Results

### 3.1. Analysis of Variance 

Various wheat traits such as grain yield, whole-kernel, flour, dough and starch quality parameters—30 in total—were compared between two waxy and two non-waxy winter wheat cultivars grown under low-input and intensive farming systems. The statistical analysis of variance (Appendix A) shows that the wheat genotype was a primary source of variation in N use efficiency, fertilizer N used for grain, grain hardness (PSI), flour sedimentation, falling number, flour yield, flour water absorption, dough development time (DDT), starch amylose content and for most viscosity characteristics. All these differences in parameters mostly represent the differences between waxy and non-waxy cultivars (which we will examine in more detail in the following steps). The amount of grain phosphorus and gluten quantity did not differ significantly between types of wheat.

Different year conditions resulted in the largest variation in test weight, protein, gluten, A-type and whole starch levels among all the factors and caused a significant effect on the changes in starch viscosity. 

The effect of the farming system was less pronounced. The great influence of the growing intensity factor was found on the variations of grain yield, N yield, P content in grains and test weight. Farming system factor did not affect N use efficiency, grain protein content, gluten quality and gluten water absorption and the majority of the flour, dough and starch quality characteristics. 

The variation of yield and quality characteristics were influenced by the interaction of factors as well. Interaction between genotypes and different growing intensity levels affected on majority of grain, flour/dough characteristics, but were not significant in determining winter wheat grain yield and starch characteristics. 

Different variations of analyzed characteristics in the factors—genotype, year or growing intensity—and their interactions demonstrated significant differences between studied genotypes in terms of wheat type. Waxy cultivars had higher variability in grain yield and gluten characteristics, PSI, A-type starch and flour/dough quality, but were less dependent in grain test weight, protein content, sedimentation and falling number characteristics. Farming systems (calculated proportion of variance for trial year and farming treatment) had significant differences between studied wheat types—waxy cultivars had higher variability in protein, gluten water absorption, flour yield, dough development time and A-type starch.

### 3.2. Effect of Year and Farming System on Grain Yield and Nitrogen (Protein) Distribution of Different Types of Winter Wheat

As shown in the analysis of variance, grain yield was significantly (*p* ≤ 0.01) influenced by growing intensity level, winter wheat type, cultivar and year (Appendix A). Intensive farming in comparison to the low-input farming system increased grain yield during the predominant stressful dry years by approximately 2.1 t ha^−1^ on average (Figure 2A). The studied cultivars were arranged in the following order by their yield performance: ‘Skagen’, ‘Suleva DS’, wx ‘Eldija’ and wx ‘Sarta’. Yield of ‘Sarta’ was significantly lower compared to the control cultivars across all trials, while yield of ‘Eldija’ was significantly lower only under high-input conditions compared to ‘Skagen’. More favorable weather conditions in 2018 resulted in a yield increase of approximately 0.6 t ha^−1^. Increased rate of nitrogen and favorable meteorological conditions caused similar improvement in yield performance for both cultivar types. However, the yield of non-waxy cultivars was more responsive to the intensive cultivation compared with low-input (the yield of non-waxy cultivars increased by 2.2–2.3 t ha^−1^, while wx cultivars only produced 1.8–1.9 t ha^−1^ yield increases).

The efficiency of N application in winter wheat grain production is an important indicator for rational N fertilization. In our research, N use efficiency (NUE) in grain production ranged from 24 to 40% (Figure 2B). Genotype and year significantly (*p* ≤ 0.01) influenced NUE. As in the yield, NUE of ‘Sarta’ was significantly lower compared to the control cultivars across all trials (except 2018 I), while that of ‘Eldija’ was significantly lower only under ‘Skagen’ in 2019. NUE tended to decrease with increasing N fertilization levels under more favorable weather conditions of 2018. 

Grain protein content was more dependent on the year than on the other two factors (Figure 2C). The yield in 2019 was slightly lower, but weather conditions were very favorable for grain quality. Presumably, the protein increase could be affected by heat waves during the 2019 grain-filling period. The differences in protein content between cultivars were mostly insignificant under both farming systems. The protein content tended to be higher under intensive growing conditions in the non-waxy cultivars, while in the wx cultivars, the protein content did not increase in 2018 and even slightly decreased in 2019. 

Grain N uptake mainly reflected the yield performance with slight corrections due to the nitrogen grain (Figure 2D). The nitrogen fertilizer percent used for grain production revealed that wx ‘Sarta’ was the most stable among cultivars but used only 51–56% N of fertilizers for the N grain yield, while other cultivars used 60–87% (Figure 2E). According to the nitrogen percent used for grain data, there was no significant effect of fertilization on separate cultivars. 

### 3.3. Effect of Year and Farming System on Grain Chemical Composition and Quality Characteristics of Different Types of Winter Wheat 

Our data show that genotype, experimental year and production system significantly affected chemical composition and quality of winter wheat grain (Appendix A and Table 1). 

Cultivar ‘Suleva DS’ showed the highest grain test weight between cultivars in the intensive production system. The 2019 year was less productive for grain test weight. In that year, the mass per hectoliter was not only lower than in previous years, it was also lower in the waxy cultivars and in the low-input production system.

Particle size index (PSI) as indicator of wheat milling and baking performance associated with wheat kernel hardness and flour granularity. In our study, higher values for PSI indicated lower hardness or softer texture of wx type grains, compared with non-waxy cultivars. Moreover wx ‘Sarta’ mostly had significantly softer grains, than wx ‘Eldija’. Grains of the 2019 harvest were significantly harder.

The intensive farming system had a tendency to decrease P content in grains compared to the low-input farming system. As grain yield increases, the amount of phosphorus could be diluted within the greater mass of grain.

Typically, wx wheat cultivars had low 62–65 s falling number (FN) values, while in regular cultivars, FN ranged from 384 to 483 s. In cultivars with typical amylose content, the values for falling number were affected by the interactions of meteorological conditions and intensity of farming. Due to meteorological conditions in the first experimental year, FN values from low-input farming tended to decrease (440 s − 54 s), while in the second year, increased (459 s + 23 s), compared to intensive. 

There were no significant differences between gluten content of waxy and non-waxy cultivars and the second year was more favorable for gluten accumulation. The gluten of the wx cultivars can absorb more water, but this tendency was observed only in one year.

Results obtained by sedimentation method for estimation of the quantity and quality of wheat proteins showed significantly lower wx wheat values (32–41 mL) than in non-waxy (44–68.5 mL) cultivars. The higher values were determined in the second year. Changes between different farming intensity levels depended on year too: lower sedimentation values were obtained using the low-input farming system in the first year, but slightly higher in the second year.

Another parameter related to the protein quality is gluten index (GI). GI values between 60–95% are considered acceptable for forming dough with good strength [28]. GI data confirmed that wx cultivars could have slightly diminished gluten strength, compared to non-waxy cultivars. GI values for wx ‘Eldija’ varied from moderate to strong gluten (GI 65–88), while those of wx ‘Sarta’, ‘Suleva DS’ and ‘Skagen’ varied from strong to very strong (GI 82–93, 83–96, 86–97). 

### 3.4. Effect of Year and Farming System on Flour and Dough Quality Characteristics of Different Types of Winter Wheat 

As presented in Appendix A, the type of wheat and cultivar is a predominant factor for flour quality variance and primary reason for differences in dough characteristics. Harvest year effect on flour and dough was also noticed, mostly as interactions with farming systems or genotype.

Waxy cultivars had significantly lower flour extraction and higher flour water absorption (FWA) with remarkable differences between cultivars (Table 2). Lowest flour extraction (50.6%) and highest absorption (73.9%) was recorded for wx ‘Eldija’, while wx ‘Sarta’ demonstrated higher (56.7%) extraction and lower (69.0%) absorption. Non-waxy cultivars by these parameters were essentially similar (with 69.1% extraction and 59.1% FWA). Wx cultivars were different by prevailing dough mixing stability time (DST) data—‘Eldija’ (with two non-wx cultivars) categorized by stability as strong flour (8 min), ‘Sarta’ as medium strong (5 min). Dough development time (DDT) also indicates the relative strength of wheat flour and can reflect the level of water absorption. Wx ‘Eldija’ is unique by other dough mixing properties as well: approximately 2.5 times longer dough development time (DDT), markedly lesser softening (DS10) and higher farinograph quality number (FQN), compared to the other three (wx/non-wx) cultivars, which according to these indicators, were just slightly different from each other. Subsequently, wx ‘Eldija’ could be more valuable than another wx cultivar ‘Sarta’ in flour mixtures and partially-processed products and could be used as an improver, not only for amylopectin-based properties. Dough mixing profiles for non-waxy and waxy winter wheat cultivars grown under different growing intensity levels are presented in Figure 3.

The typical and most significant differences of dough mixing parameters were established between low-input and intensive farming systems just in the first year (2018). Low-input system dough quality was often markedly poorer compared with that from high-input wheat growing. In the second year (2019), the flour yield and dough quality parameters were substantially better; the differences in dough among both farming systems were insignificant or better in low-input farming system. 

### 3.5. Effect of Year and Farming System on Different Type Winter Wheat Starch Quality Characteristics

According to the analysis of variance (Appendix A), farming intensity level did not significantly affect starch quality characteristics (with exception of starch and A-starch parameters); therefore, this factor was not analyzed in detail (Table 3). The highest whole starch and A-type starch variations were associated with year conditions, while amylose content (~0.52% in wx, and 26.2% in non-wx cultivars) and starch viscosity distribution clearly depended on the wheat type but were also influenced by year conditions (Table 3, Figure 4A,B). A longer heat period during grain filling time in 2019 caused not only higher grain protein content, but also lower starch content (by 2.1% unit in average), higher volume of A-type granule accumulation (especially in earlier wx cultivars) and slightly higher amylose content (in non-wx cultivars) (Figure 4A). Furthermore, the 2018 was associated with more conducive A-type starch accumulation in wx cultivars (especially ‘Eldija’) when they were grown at higher intensity. While in 2019, under the influence of meteorological conditions, whole starch content was lower under low-input farming system.

Specific properties of waxy wheat starches such as higher peak viscosity, lower retrogradation rate (setback) and higher digestibility (due to the predominant amylopectin) are desired for a range of applications. In our experiment, peak viscosity in wx cultivars (compared with non-wx) was greater by 97 RVU; viscosity in setback was lower by 77 RVU (Table 3). Furthermore, waxy starches attributed to the lower gelatinization temperatures, rapid and more complete hydrolyzation than non-waxy starches (in our case, approximately by 15 °C and 3 min). 

The influence of meteorological conditions on starch pasting properties (Table 3) of non-waxy and waxy wheat cultivars was also evident. Pasting characteristics of wheat starches during cycles of heating and cooling are presented in Figure 5. In 2018, wx ‘Eldija’ demonstrated higher through and final viscosities in comparison with wx ‘Sarta’. In the next year, ‘Skagen’ demonstrated increased through viscosity in comparison with ‘Suleva DS’. In our experiment, due to the meteorological conditions in 2019, viscosity on average increased from 47 RVU at peak to 72 RVU at final, compared to 2018. It was characteristic for both wheat types.

### 3.6. PCA Factor Loadings Based on Correlations

Principal component analysis (PCA) was conducted to obtain an overall visualization of multivariate interactions. The ordination of wheat type, cultivar, year and farming system treatments using all analyzed yield and quality variables on the PCA biplots are depicted in Figure 6. PCA loadings based on correlations are presented in Table 4. The most informative first four principal components explained 85.3% of the data variability. In relation to the first component axis (Figure 6A), associated with 46.3% of the variability, samples of the non-waxy wheat cultivars ordinated on the left side of the biplot, while samples of the waxy wheat cultivars ordinated on the right side. Therefore, the relationships between PC1 and variables can be interpreted as associations with wheat type.

In relation to the first principal component (PC1) analysis of wheat types, a low amylose content was strongly correlated (*r* ≥ 0.8) with lower falling number, flour yield and sedimentation values, lower nitrogen % used for grain, higher flour water absorption and PSI, shorter starch peak viscosity time, lower viscosity temperature, higher peak and breakdown viscosity, but lower setback and final viscosity values. Additionally, low amylose content (associated with waxy wheat) was moderately (*r* ≥ 0.5÷0.6) correlated with lower grain yield, N uptake, starch content, gluten index and higher dough development time (DDT) values. DDT increase (as analyzed in Table 3) was due to only one wx cultivar ‘Eldija’.

In relation to the second principal component (PC2), which accounted for 20% of the variance, the samples of 2018 were separated from the 2019 samples (Figure 6B, Table 4). In this clustering, related with meteorological conditions, decreasing protein content data is associated with lower but stronger gluten, shorter dough development time, higher dough softening, higher starch content, lower A-starch, peak and final viscosity values but higher grain test weight. It is likely that prevailing heat waves during the grain filling period in 2019 decreased grain test weight but increased protein and gluten content, though likely also caused gluten to be weaker. Dough development time became longer, dough softening lower and starch content decreased, but A-starch, starch peak and final viscosity values increased. 

The third principal component (PC3) accounted for 10.6% of variability associated with cultivar peculiarities in dough mixing (Figure 6C, Table 4). By this clustering, distribution of wx ‘Eldija’ samples was closer to the non-waxy cultivars, compared to that of another wx cultivar ‘Sarta’. In PC3, decreasing dough stability related to shorter dough development and higher softening, lower gluten water absorption and lower N use efficiency. 

The fourth principal component (PC4) accounts for 7.9% of the variance and shows clustering by growing intensity levels (Figure 6D, Table 4). In this part, N unavailability had a strong (or moderate) negative effect on grain yield, grain N uptake and grain test weight but increased P content in grain. 

The quality of the dough was assessed by rheological properties and the results showed that it depended on a combination of several factors, in particular, on the gluten characteristics, which in turn depended on the studied trial factors (weather conditions in trial years and varieties, but not farming intensity). 

## 4. Discussion

According to our experiment, the yield performance of the waxy wheat cultivar ‘Sarta’ was significantly lower than that of two non-waxy cultivars in two different intensity farming systems over two years. The yield of another waxy cultivar ‘Eldija’ was lower than that of two non-waxy cultivars, but the differences were mainly not significant (Figure 2). The lower yield performance of waxy wheats ‘Sarta’ and ‘Eldija’ can be explained by the differences in vegetation duration, as earlier genotypes have lower grain yield. A significantly shorter vegetation period indicates that they did not inherit the proper combination of photoperiod sensitivity alleles and earliness genes for environments in Lithuania. Furthermore, the modification of the starch profile that comprises a major part of the endosperm could cause inevitable consequences on the agronomic performance and quality parameters. The waxy wheat has been studied for more than 20 years but the effect of Wx genes on yield performance and quality characteristics is still not completely clear. The published results concerning the association between Wx genes, yield and grain quality traits are inconsistent and sometimes contradictory [10,29]. Several studies have been performed comparing waxy and non-waxy genotypes, which do not share common genetic backgrounds [8,10,22,30], while accurate determination on the true genetic effects of the null alleles are possible if they are compared in the same genetic background, which differs only by these alleles. Recently, Zi et al. [31] compared sucrose conversion to the starch of waxy and non-waxy cultivars and found that starch synthesis of waxy wheat is weaker in the late grain filling stage. The authors suggested that the absence of GBSS enzyme confers a complementary effect on other responsible enzymes for starch biosynthesis and eventually causes the lower total starch content in grains. However, the comparison of starch synthesis abilities was carried out on different genetic backgrounds. According to our experiment, the total starch content of waxy cultivars was slightly lower (from 1.45 to 1.95% lower on average) in comparison to the control cultivar ‘Suleva DS’; however, the differences were not significant (Figure 4). Several studies have been conducted to date on waxy genes by using near-isogenic lines. For instance, Miura et al. [23] did not find significant differences in yield performance between isogenic lines of different Wx groups. However, the yield performance was measured from single-meter rows in one season and one location [23]; that cannot provide the precise evaluation of the influence of null alleles on the yield. Vignaux et al. [25] examined the recombinant inbred lines (isogenic lines) with different variants of waxy genes in plots of around 6 m^2^. Their report showed that the presence of two waxy genes *Wx-A1* and *Wx-B1* did not significantly affect either the yield performance, kernel size or kernel hardness of durum wheat lines at two locations in North Dakota [25]. Hucl and Ramachandran [29] studied 32 near isogenic lines (NILs) in plots of 3.6 m^2^ at two locations for three years. According to their study, isogenic lines with three waxy genes demonstrated similar yield and thousand kernel weight compared to the non-waxy isogenic line. Surprisingly, partially waxy NILs of durum wheat even exhibited slightly higher yield and thousand kernel weight in comparison to wild type isogenic lines [29]. The lower performance of the studied waxy cultivars ‘Eldija’ and ‘Sarta’ might be explained by their pedigree and relatively short breeding time. Two waxy lines provided by Graybosch were selected as they were more adapted and were used in crossings. Afterwards, more adapted waxy lines to our environments were developed from this cross. ‘Eldija’ and ‘Sarta’ were developed by crossing locally adapted cultivars with our waxy line developed from Graybosch lines. Because there were only two rounds of crossing, it was not possible to combine all appropriate alleles, since the development of highly adapted and productive waxy cultivars requires several rounds of crossing and selection. To date, only few cultivars have been developed and recommended for growing in the European countries: cv. ‘Waxydie’ was developed in Germany, cv. ‘Waximum’ in France and cv. ‘Minija DS’ in Lithuania. The amylose content of their grain is below 1%, but yield performance is inferior compared to that of the non-waxy cultivars [32,33,34]. However, the waxy wheat cultivar ‘Waximum’ produced 1.6% higher yield under fungicide treatment compared to the average values for the four non-waxy standarts and a slight yield decrease of −0.3% without the treatment [34]. Regardless of the benefits of waxy starch for some food and non-food industries and the fact that yield performance can evidently be improved, such small number of registered waxy wheat cultivars demonstrates that waxy wheat is so far a new type of wheat and largely not adapted to the environments of Europe. Therefore, introgression of waxy genes into the elite breeding material and elimination of undesirable traits of initial parents by conventional breeding methods is a challenging task for wheat breeders.

Another discussion point is the association of waxy type with quality parameters. Apparently, such characteristics as poor flour yield and low falling number values is a natural consequence of modified starch profile. Rheological dough properties can be affected both by protein and starch fraction. In our study, the protein quality of waxy cultivars measured by sedimentation volume was significantly lower under two intensity farming systems during a two-year crop season. Another quality parameter, gluten index of waxy cultivars, was also lower in almost all trials (Table 1). However, the study on the isolated gluten fraction of waxy wheat did not demonstrate any significant differences in terms of loaf volume and crumb grain scores [35]. It should be mentioned that Sayaslan et al. [35] used the starch–stress bake test of Miller and Hoseney [36], according to which the samples contained about 50% of waxy wheat gluten after mixing with wheat flour; therefore, it could not demonstrate a complete influence of waxy gluten [35]. Graybosch et al. [10] analyzed the grain quality characteristics of waxy and non-waxy wheat in the different genetic backgrounds and did not compare the average values but demonstrated variation inside waxy genotype groups. According to their study, about 50% of the waxy lines did not differ significantly from the highest quality non-waxy cultivar for protein content and 44% of the waxy lines in the gluten index. That might indicate that waxy cultivars with strong gluten quality can be developed. The weaker gluten quality of ‘Eldija’ and ‘Sarta’ might be caused not by Wx genes, but rather by other factors of genetic backgrounds, which are unlikely to be linked with Wx genes and can be eliminated through the breeding cycles. However, the changed amylose-to-amylopectin ratio undoubtedly affects the dough quality of wheat. The crystalline pattern of waxy starch granules is different from that of non-waxy wheat, which inevitably influences the texture, stability and viscosity of the dough. In our study, dough parameters such as peak viscosity, trough, breakdown, setback, peak time and viscosity temperature, which depend on the starch profile significantly differed from those of non-waxy cultivars, which are in line with the results of previous studies [37,38,39].

According to our study the three factors (genotype, year conditions and intensity of growing) affected yield and grain characteristics of both types of winter wheat in a similar way. The results indicate that studied waxy cultivars (‘Sarta’ and ‘Eldija’) demonstrated relatively stable yield performance compared with two standart cultivars under two different farming intensity levels, for the two cropping seasons. Analysis of variance demonstrated significant differences between waxy and non-waxy cultivars in terms of falling number, gluten index, flour yield and amylose content (Appendix A). This can be explained by the difference in amylose-to-amylopectin ratio of the starch fraction [22]. 

The particular characteristics of waxy wheat starches, such as greater viscosities (peak viscosity), low retrogradation rate (setback) and high digestibility (due to the predominant amylopectin), present unique and advantageous properties for a range of industries [40]. Waxy starches are characterized by lower gelatinization temperatures, quicker hydrolyzation and more efficient conversion to sugar compared to the normal starches. This makes it preferable for the development of starch-derived sweeteners and industrial alcohol. Based on the property of a lower starch setback in waxy wheat, which is also confirmed in our research, the retrogradation speed of frozen food can be reduced and the shelf-life of bread products can be prolonged by blending waxy wheat flour with conventional flour [40,41].

The associations among protein, gluten content, quality properties of gluten and environments have been studied for many years. In contrast with the many previous studies [42,43], our findings demonstrated much weaker G × Y, G × F, Y × F, G × Y × F (G—genotype, Y—year, F—farming system) interactions for protein and gluten contents and gluten index. However, the significant and strong G × F interactions were found for sedimentation values for both types of wheat. The contradiction between our results and those obtained in other experiments might be explained by the different environmental conditions, farming systems and/or genotypes. Most of the previous studies have demonstrated that protein content is lower in the grain grown under organic and low-input farming systems compared to the conventional high-input farming [43,44,45]. Interestingly, protein content of the studied waxy cultivars grown under intensive farming remained at the same levels under low-input farming for two consecutive years in our study (Figure 2). Similarly, Mäder et al. [46] did not find differences in the protein content between non-waxy wheat cultivars grown under low and high-input farming. Heat waves and the lack of precipitation lead to the weaker synthesis of starch resulting in the increased proportion of A-type starch and lower yield performance. At the same time, it can cause the increase in protein content, especially when the heat period occurs at the beginning of grain filling [47,48,49]. This might explain a reduction in yield and increase in protein content in our experiment in 2019. Our findings demonstrate that the prevailing heat waves during the grain filling period decreased grain test weight values, increased protein and gluten content accumulation and caused weaker gluten formation. Dough development time under these conditions became longer, dough softening lowered and accumulation of starch content decreased; however, the volume of A-starch and values for starch viscosities (at peak and final) increased.

Sufficient NPK fertilization is a necessary requirement for modern wheat cultivars to produce high grain yields with appropriate quality in order to meet the requirements of bakeries and other food industries. It is well known that crop yield performance and grain quality are directly associated with the rate of nitrogen fertilization. The lowering of nitrogen input can cause not only the reduction in grain yield but also poor grain quality, and as a result, lower wheat grain class. However, higher input of fertilizer than can be assimilated by the crop leads to a leak of fertilizer through the runoff water and consequently to a lower nitrogen use efficiency. The general nitrogen use efficiency for all grain crops is approximately 33% [50]. This means that about 67% of the applied nitrogen can be lost and pollute the environment. Therefore, improvement in nitrogen use efficiency is critical for sustainable agriculture [51,52]. The nitrogen use efficiency (NUE) depends on the rate of N fertilizer, weather conditions, amount and frequency of precipitation [53]. During the two years of our experiment, grain yield stability and response to different nitrogen rates of the two waxy cultivars were similar to those of the non-waxy cultivars. The nitrogen use efficiency of waxy cultivars was reduced similarly to the yield performance (Figure 2). Mean NUE of waxy cultivars was lower compared to the non-waxy cultivars, but in most cases, the NUE of the waxy cultivar ‘Eldija’ was not significantly lower than that of the two non-waxy cultivars (Figure 2). In general, the NUE was higher for all the studied cultivars under more favorable weather conditions in 2018. Apart from these factors, the results indicate that the increased nitrogen rate (N_200_) did not significantly affect the grain quality characteristics (sedimentation and gluten index) of waxy cultivars in two consecutive growing seasons. At the same time, it negatively affected content of phosphorus in the grain.

In sum, the waxy breeding program began in Lithuania relatively recently. Lithuanian waxy cultivars have been developed to meet a potential demand and demonstrate the availability of new wheat starch for industrial applications [33]. Waxy cultivars differ from non-waxy starch cultivars not only by low amylose content and low falling number values, they often have lower N use efficiency. The grains of waxy cultivars are characterized by softer texture and have lower sedimentation values and lower flour yield, but higher flour water absorption. Furthermore, waxy starches reach gelatinization approximately 3 min quicker and under temperatures lower by 15 °C. Moreover, they were characterized by the greater peak viscosity and lower through final and retrogradation (setback) viscosities. The waxy cultivar ‘Eldija’ is unique by its flour absorption and dough mixing properties: it has significantly higher water absorption (the flour on average absorbed 73.8% water), approximately 2.5 times longer dough development time, significantly lower dough softening and a higher farinograph quality number. Moreover, comparison of dough quality parameters between wx ‘Eldija’ and wx ‘Sarta’ and non-waxy cultivars demonstrated that wx ‘Eldija’ is more valued than another wx cultivar ‘Sarta’ for use in flour mixtures and partially-processed products, and could be used as an improver not only for amylopectin-based properties but also for other dough properties. The main agronomic traits of the waxy cultivars demonstrated similar plasticity in terms of year and cultivation intensity effects. Protein and gluten content of the waxy cultivars exhibited even higher stability under different intensity farming systems for two consecutive years. Several quality traits of waxy cultivars such as falling number, flour yield and dough mixing properties were inferior to those of the non-waxy cultivars. These properties are difficult to improve as they are tightly associated with the waxy starch trait. In contrast, yield performance and gluten quality are more amenable for improvement. However, the breeding of waxy wheat is hampered by a limited pool of waxy wheats and, therefore, substantial breeding efforts are needed to develop highly competitive cultivars with a new starch profile to meet the preferences of wheat growers and customers.

## Figures and Tables

**Figure 1 plants-11-00882-f001:**
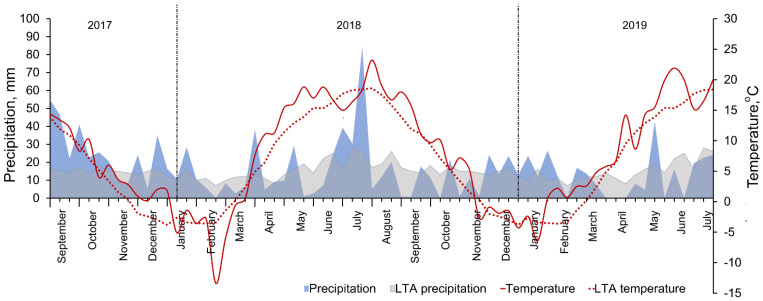
Meteorological conditions—precipitation and air temperature—during winter wheat growing seasons of 2017–2019 and long-term average (LTA, 1924–2019). (Dotnuva Meteorological Station data).

**Figure 2 plants-11-00882-f002:**
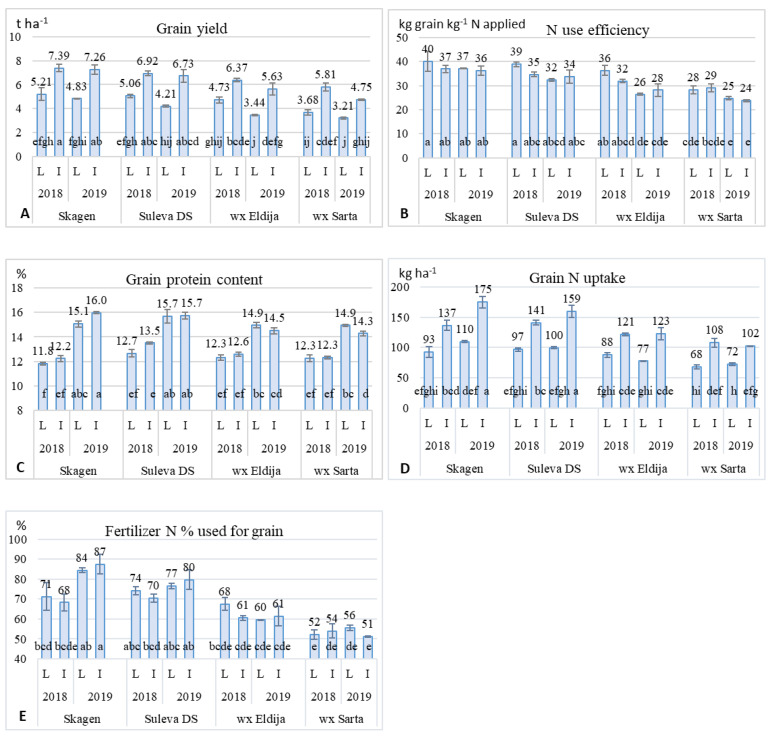
Wheat grain yield (**A**), N use efficiency (**B**), protein (**C**), N uptake (**D**) and fertilizer N used for grain (**E**) distribution through non-waxy (‘Skagen’, ‘Suleva DS’) and waxy (‘Eldija’, ‘Sarta’) winter wheat cultivars grown under different farming systems (L—low-input, I—intensive) in two cropping seasons. Values in columns followed by the same letter are not significantly different at *p* < 0.05.

**Figure 3 plants-11-00882-f003:**
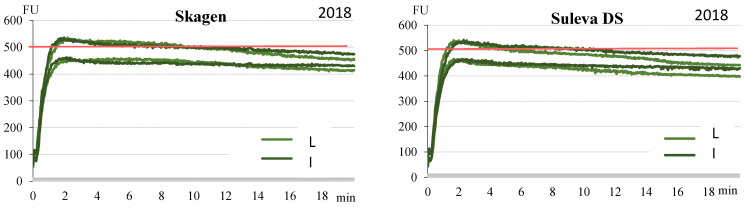
Dough mixing profiles for non-waxy (‘Skagen’, ‘Suleva DS’) and waxy (‘Eldija’, ‘Sarta’) winter wheat cultivars grown under different farming systems (L—low-input, I—intensive) in 2018. Analysis performed with Brabender’s farinograph.

**Figure 4 plants-11-00882-f004:**
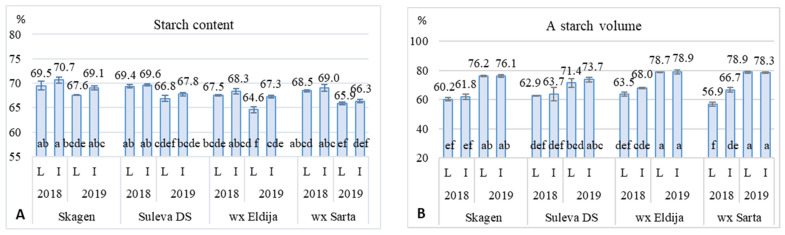
Grain starch content (**A**) and A-type starch volume (**B**) of non-waxy (‘Skagen’, ‘Suleva DS’) and waxy (‘Eldija’, ‘Sarta’) winter wheat cultivars grown under different farming systems (L—low-input, I—intensive) in two cropping seasons. Values in columns followed by the same letter are not significantly different at *p* < 0.05.

**Figure 5 plants-11-00882-f005:**
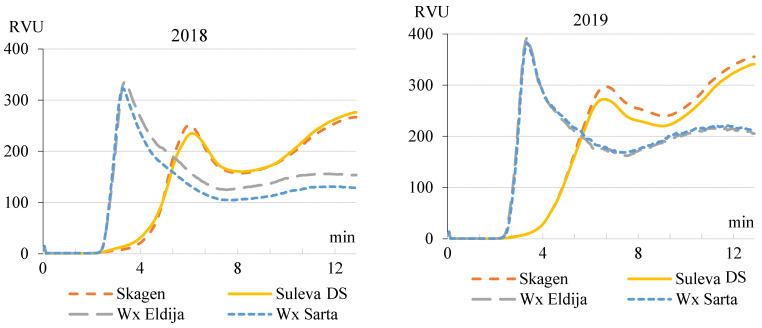
Starch pasting profiles for non-waxy (‘Skagen’, ‘Suleva DS’) and waxy (‘Eldija’, ‘Sarta’) wheat cultivars grown over two cropping seasons (averaged data of two farming systems). Analysis performed with RVA—rapid visco analyzer; RVU—rapid visco units.

**Figure 6 plants-11-00882-f006:**
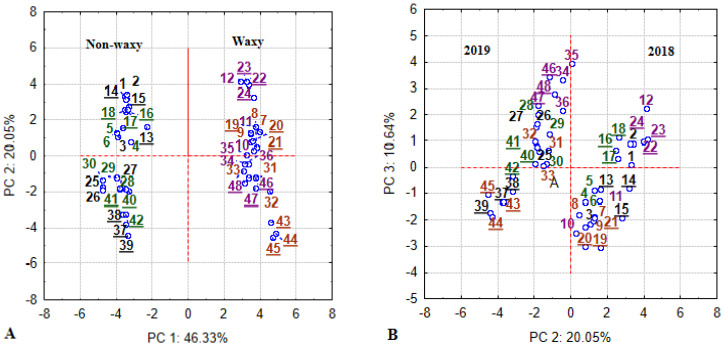
Loading plots from principal components analysis (PCA): PC1 × PC2 scores plot (**A**), PC2 × PC3 scores plot (**B**), PC3 × PC4 scores plot (**C**), PC4 × PC5 scores plot (**D**). Analysis was conducted using data of grain yield and quality characteristics when 4 winter wheat cultivars (2 non-waxy and 2 waxy) grown in low-input and intensive farming systems over 2-year conditions. Designation: black—‘Skagen’, green—‘Suleva DS’, brown—Wx ‘Eldija’, purple—Wx ‘Sarta’, not underlined—low-input, underlined—intensive, No 1–24—2018, No 25–48—2019.

**Table 1 plants-11-00882-t001:** Grain chemical composition and quality characteristics of non-waxy and waxy winter wheat cultivars grown under different farming systems in two cropping seasons.

Wheat Type /Cultivar	Year	Farming System	Test Weight g L^−1^	Particle Size Index	P, %	Falling Number, s	Sedimen-tation, mL	Wet Gluten Content, %	Gluten Index, %	Gluten H_2_O Absorption, %
Non-waxy cultivars										
Skagen	2018	L	791bc	5.3g	0.32abcd	439c	44.0c	21.7h	96.5a	64.6cde
		I	798b	5.0g	0.27cd	388d	56.7b	22.0h	97.4a	62.7e
	2019	L	738g	6.7f	0.38a	451bc	68.5a	30.6a	94.6abc	65.7bcd
		I	791bc	7.8e	0.27cd	476a	67.7a	29.9ab	86.3abcd	64.3cde
Suleva DS	2018	L	788c	5.1g	0.33abc	441c	51.3b	22.7gh	95.2ab	64.1cde
		I	814a	5.4g	0.28bcd	384d	65.3a	25.1def	96.3a	64.4cde
	2019	L	779d	6.7f	0.29bcd	467ab	66.7a	29.6ab	93.4abcd	64.4cde
		I	809a	7.2ef	0.29bcd	483a	65.0a	29.1ab	83.4cd	64.3cde
Waxy cultivars										
Eldija	2018	L	790c	8.4d	0.34abc	64.0e	32.3e	24.9defg	85.6bcd	68.3a
		I	795bc	9.0c	0.29bcd	65.0e	34.5de	24.9defg	87.9abcd	66.2abc
	2019	L	748f	9.5b	0.32abcd	64.7e	40.8cd	28.1abc	67.8e	63.7de
		I	772d	9.8ab	0.27cd	65.3e	36.5de	27.5bcd	65.4e	65.5bcd
Sarta	2018	L	792bc	9.1bc	0.32abcd	62.0e	34.2e	24.3fgh	91.2abcd	67.5ab
		I	795bc	9.6ab	0.28bcd	62.7e	34.7de	24.6efgh	92.7abcd	65.1cde
	2019	L	735g	10.0ab	0.35ab	62.0e	40.5d	26.8cde	81.5d	63.5de
		I	757e	10.5a	0.24d	62.0e	36.7de	25.6cdef	82.9d	63.4de

L—low-input, I—intensive. Values in the column followed by the same letter are not significantly different at *p* < 0.05.

**Table 2 plants-11-00882-t002:** Flour and dough quality characteristics of non-waxy and waxy winter wheat cultivars grown under different farming systems in two cropping seasons.

Wheat Type /Cultivar	Year	Farming System	Flour Yield, %	Farinograph Analysis					
				FWA, mL	DDT, min.	DST, min.	DS10, FU	DS12, FU	FQN
Non-waxy cultivars									
Skagen	2018	L	69.7a	59.3cde	2.5fgh	9.2a	31abcd	47abcd	100bcd
		I	63.7bc	60.2c	2.2h	7.5a	34abc	42bcd	76d
	2019	L	71.2a	58.4de	6.0cde	8.9a	12cd	26de	187a
		I	71.9a	58.0e	5.4def	4.4a	30abcd	62abc	97bcd
Suleva DS	2018	L	68.3ab	59.5cde	2.2h	5.0a	45ab	62abc	57d
		I	67.3ab	59.9cd	2.5fgh	10.2a	27abcd	39cd	111bcd
	2019	L	69.3a	58.8cde	3.4efgh	9.5a	21bcd	36cd	128abcd
		I	71.2a	58.8cde	4.2defgh	5.3a	31abcd	56abc	100bcd
Waxy cultivars									
Eldija	2018	L	48.6g	74.2a	8.6bc	8.1a	7d	67a	148abc
		I	48.8g	74.3a	10.1ab	7.2a	7d	63abc	161ab
	2019	L	48.7g	73.4a	12.1a	10.2a	5d	0e	194a
		I	56.6de	73.3a	7.0cd	6.2a	11cd	57abc	126abcd
Sarta	2018	L	56.0ef	69.3b	2.4gh	4.7a	48a	69a	69d
		I	51.1fg	69.2b	5.5def	8.1a	26bcd	49abcd	124abcd
	2019	L	58.2de	69.3b	3.8efgh	4.2a	35abc	57abc	93bcd
		I	61.6cd	68.1b	2.9fgh	5.0a	36abc	66ab	85cd

L—low-input, I—intensive, FWA—flour water absorption, DDT—dough development time, DST—dough stability time, DS10—degree of softening after 10 min after mixing starts, DS12—degree of softening at 12 min. after dough mixing peak, FQN—farinograph quality number, FU—farinograph units. Values in the column followed by the same letter are not significantly different at *p* < 0.05.

**Table 3 plants-11-00882-t003:** Starch quality characteristics of non-waxy and waxy winter wheat cultivars grown in two cropping seasons (averaged data of two farming systems).

Cultivar	Year	Amylose, %	RVA Analysis						
			Peak Visc., RVU	Trough Visc., RVU	Breakdown Visc., RVU	Final Visc., RVU	Setback Visc., RVU	Peak Time, min.	Visc. Temp., °C
Skagen	2018	25.3b	252f	157b	95b	267b	110a	6.1b	85.2a
Suleva DS		25.2b	240f	160b	80bc	276b	116a	6.2b	82.4a
Wx Eldija		0.70c	344bc	124c	220a	154d	29c	3.5c	67.8b
Wx Sarta		0.00c	325cd	103d	223a	129e	26c	3.4c	67.8b
Skagen	2019	26.8a	297de	235a	62bc	355a	120a	6.8a	82.4a
Suleva DS		27.4a	273ef	220a	53c	341a	121a	6.7a	82.0a
Wx Eldija		0.83c	396a	157b	238a	206c	48b	3.4c	67.4b
Wx Sarta		0.55c	384ab	160b	224a	214c	54b	3.4c	67.7b

Visc.—viscosity; temp.—temperature; RVA—rapid viscosity analyzer; RVU—rapid viscosity analyzer units. Values in the column followed by the same letter are not significantly different at *p* < 0.05.

**Table 4 plants-11-00882-t004:** Correlations between the first four principal components and 30 winter wheat grain yield and quality traits for calculation representing 2 waxy and 2 non-waxy winter wheat cultivars grown in low-input and intensive farming systems under 2-year conditions (n = 48).

Active and Supplementary * Variable	PC 1	PC 2	PC 3	PC 4
	Wheat type	Year	Cultivar peculiarities	Farming system
	Total variance %			
	46.3	20.1	10.6	7.9
	Correlation			
Grain yield	**−0.520**	0.189	−0.222	**−0.758**
N use efficiency	**−0.694**	0.233	**−0.497**	−0.022
Grain N yield (uptake)	**−0.579**	−0.144	−0.050	**−0.775**
Fertilizer N% used for grain	**−0.802**	−0.302	−0.258	−0.113
Grain protein content	−0.204	**−0.859**	0.359	−0.111
Test weight	−0.368	**0.598**	−0.267	**−0.494**
Particle size index	**0.854**	−0.270	0.291	−0.190
Phosphorus content	−0.013	−0.111	−0.242	**0.703**
Sedimentation	**−0.886**	−0.370	0.022	−0.022
Wet gluten content	−0.102	**−0.863**	0.133	−0.103
Gluten index	**−0.575**	**0.507**	−0.190	0.202
Gluten H_2_O absorption	0.326	0.182	**−0.429**	0.037
Falling number	**−0.985**	−0.086	−0.023	0.070
Flour yield	**−0.903**	−0.101	0.300	0.085
Flour water absorption (BF)	**0.965**	0.005	−0.152	−0.126
Dough development time (BF)	**0.523**	**−0.505**	**−0.529**	−0.197
Dough stability time (BF)	−0.076	−0.261	**−0.745**	0.152
Degree of softening10 (BF)	−0.278	**0.561**	**0.691**	0.165
Degree of softening12 (BF)	0.111	**0.550**	**0.424**	−0.278
BF quality number (BF)	0.263	**−0.650**	**−0.669**	0.037
Starch content	**−0.525**	**0.688**	−0.148	−0.216
A-type starch	0.187	**−0.841**	0.329	−0.172
Amylose content	**−0.991**	−0.063	−0.044	0.059
Peak viscosity (RVA)	**0.830**	**−0.414**	0.094	−0.161
Trough (RVA)	**−0.681**	**−0.666**	0.172	−0.038
Breakdown (RVA)	**0.969**	0.051	−0.023	−0.098
Final viscosity (RVA)	**−0.863**	**−0.467**	0.122	0.009
Setback (RVA)	**−0.950**	−0.201	0.053	0.058
Peak Time (RVA)	**−0.983**	−0.117	−0.024	0.085
Viscosity temperature (RVA)	**−0.936**	0.062	−0.040	0.143
Cultivar *	−0.868	−0.119	−0.273	0.003
Type of wheat *	**−0.990**	−0.025	−0.052	0.079
Year *	0.012	**0.859**	−0.456	0.042
Growing intensity *	0.068	−0.073	−0.129	**0.886**

PC—principal component, N—nitrogen; BF—Brabender’s farinograph; RVA—rapid visco analyzer. Bold text indicates correlation |*r|* ≤ −0.4 and ≥0.4. * supplementary variable.

## Data Availability

Not applicable.

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
