# Peer review of "Grain Yield Performance and Quality Characteristics of Waxy and Non-Waxy Winter Wheat Cultivars under High and Low-Input Farming Systems"

_plants, 2022, doi:10.3390/plants11070882_

Round 1
Reviewer 1 Report
The manuscript is well written, clear and it provides new scientific evidence in the field. However, there are few aspects that demand improving. Below I list remarks that shall be considered by the author:
Abstract
I wonder why the authors use both: wheat varieties and genotypes. The samples investigated in this manuscript were all wheat varieties not lines of one variety therefore, I suggest using varieties only.
The farming system was not only influenced by the nitrogen fertilization but also the use of seed treatment, plant growth and fungicides and insecticides. It is needed to provide this information in the abstract (line 10,11).
Please avoid abbreviation in the abstract and use either full wording or both full wording followed by an abbreviation in brackets (line 26).
- Introduction
It is necessary to state a research hypothesis in the introduction section.
2.2. Weather conditions
Statement by the authors that “drought period was recorded from late May to early July in 2018, and early April to early July in 2019. (line 126-127) is contradicting to the information provided in Fig. 1 where drought was in April 2019 and in June 2019. Additionally, using terms such as “late May or Early June” (line 127) should be avoided because in the manuscript (Fig. 1) only monthly precipitation was shown. Please refer to whole months and not parts of it unless additional weather data was recorded and can be provided.
- Results
In general this section is well written and focused on showing the interaction between investigated factors (years, variety, intensity of the production) and how they shaped the investigated characteristics. I do have few minor remarks below:
Line 299 The sentence is true but with the exception of wx Sarta
Line 305 – “The protein content tended to be higher under intensive growing conditions in the non-waxy cultivars”
There was no difference in 2019 so that needs to be mentioned.
Line 327 – “The highest grain test weight produced cultivar ‘Suleva’”
This is true only in intensive production system.
- References
There is a large number of citations present in the text but not featured in the reference list. The list must be supplied with the missing references detailed below.
Regina et al., 2015, Botticela et al., 2018,
Graybosch et al. 2003 - Zi et al. (2018),
Bundessortenamt, 2017, SECOBRA Recherches, 2011, Bajaj, & Kaur, 2017)., Kamran et al., 2014, Osman et al., 2012
There is a large number of citations present in the reference list but not in the manuscript.
Borkowska, H.; Grundas, S.; Styk,B 1999
Jonnala, R. S., MacRitchie, F., Smail, V. W., Seabourn, B. W., Tilley, M., Lafiandra, D., & Urbano, M. (2010).
Kim, W., chemistry, P. S.-C., & 1993,
Raun, W. R., & Johnson, G. V. (1999).
Rosicka-Kaczmarek, J., Kwaśniewska-Karolak, I., Nebesny, E., & Komisarczyk, A. (2018)
Zhu, T., Jackson, D. S., Wehling, R. L., & Geera, B. (2008).

Author Response
Dear Reviewers and Editor,
We are grateful for valuable advice on how to improve the quality of our manuscript.
We edited the manuscript regarding your recommendations, detailing our changes presented in the tables below. The latest version of the manuscript uploaded to the system has been corrected using “Track Changes”.
Thank you for your time
Regards,
The authors
10 March 2022
Reviewer 1.
Open Review
(x) I would not like to sign my review report
( ) I would like to sign my review report
English language and style
( ) Extensive editing of English language and style required
( ) Moderate English changes required
( ) English language and style are fine/minor spell check required
(x) I don't feel qualified to judge about the English language and style
|
Yes |
Can be improved |
Must be improved |
Not applicable |
|
|
Does the introduction provide sufficient background and include all relevant references? |
( ) |
(x) |
( ) |
( ) |
|
Is the research design appropriate? |
(x) |
( ) |
( ) |
( ) |
|
Are the methods adequately described? |
(x) |
( ) |
( ) |
( ) |
|
Are the results clearly presented? |
( ) |
(x) |
( ) |
( ) |
|
Are the conclusions supported by the results? |
(x) |
( ) |
( ) |
( ) |
Comments and Suggestions for Authors
The manuscript is well written, clear and it provides new scientific evidence in the field. However, there are few aspects that demand improving. Below I list remarks that shall be considered by the author:
|
Comments and suggestions for authors |
Appropriate adjustments in manuscript have been made |
|
Abstract I wonder why the authors use both: wheat varieties and genotypes. The samples investigated in this manuscript were all wheat varieties not lines of one variety therefore, I suggest using varieties only. |
As suggested, the term “genotypes” was replaced with “cultivars”. |
|
The farming system was not only influenced by the nitrogen fertilization but also the use of seed treatment, plant growth and fungicides and insecticides. It is needed to provide this information in the abstract (line 10,11). |
The abstract was reviewed and adjusted to consider the comments of all reviewers. |
|
Please avoid abbreviation in the abstract and use either full wording or both full wording followed by an abbreviation in brackets (line 26). |
As suggested, abbreviations were replaced with full words. |
|
1. Introduction It is necessary to state a research hypothesis in the introduction section. |
Hypothesis was added to the Introduction. |
|
2.2. Weather conditions Statement by the authors that “drought period was recorded from late May to early July in 2018, and early April to early July in 2019. (line 126-127) is contradicting to the information provided in Fig. 1 where drought was in April 2019 and in June 2019. Additionally, using terms such as “late May or Early June” (line 127) should be avoided because in the manuscript (Fig. 1) only monthly precipitation was shown. Please refer to whole months and not parts of it unless additional weather data was recorded and can be provided. |
We have changed the figure, now it provides more detailed data. Referred text was revised and rewritten. |
|
Results In general this section is well written and focused on showing the interaction between investigated factors (years, variety, intensity of the production) and how they shaped the investigated characteristics. I do have few minor remarks below: Line 299 The sentence is true but with the exception of wx Sarta |
Data and expression were re-checked. |
|
Line 305 – “The protein content tended to be higher under intensive growing conditions in the non-waxy cultivars” There was no difference in 2019 so that needs to be mentioned. |
Data re-checked and corrected. |
|
Line 327 – “The highest grain test weight produced cultivar ‘Suleva’. This is true only in intensive production system. |
Corrected. |
|
4. References There is a large number of citations present in the text but not featured in the reference list. The list must be supplied with the missing references detailed below. Regina et al., 2015, Botticela et al., 2018, Graybosch et al. 2003 - Zi et al. (2018), Bundessortenamt, 2017, SECOBRA Recherches, 2011, Bajaj, & Kaur, 2017)., Kamran et al., 2014, Osman et al., 2012 There is a large number of citations present in the reference list but not in the manuscript. Borkowska, H.; Grundas, S.; Styk,B 1999 Jonnala, R. S., MacRitchie, F., Smail, V. W., Seabourn, B. W., Tilley, M., Lafiandra, D., & Urbano, M. (2010). Kim, W., chemistry, P. S.-C., & 1993, Raun, W. R., & Johnson, G. V. (1999). Rosicka-Kaczmarek, J., Kwaśniewska-Karolak, I., Nebesny, E., & Komisarczyk, A. (2018) Zhu, T., Jackson, D. S., Wehling, R. L., & Geera, B. (2008). |
Omitted references were added. |
Reviewer 2 Report
The article suffers from serious linguistical errors, which must be corrected before any decision. The text is not clear in many cases and has to be written again (i. e. lines 120-122; lines 312-315; 327-330; 343-345; 380-381; 433-435; Furthermore, there are references in the text which are missing from the reference list (Regina et. Al., 2015, Botticela et. Al., 2018 both in line 41; Graybosch et al., 2003, line 522; Zi et al., 2018, line 525; Bundessortenamt, 2017 in line 560-561. On the hand, three articles in the reference list ate not referred in the text (lines 730-732; 760-762; 809-812). Another issue is that the authors do not follow the Journal’s instructions on how to write a references especially in cases that the authors are more than two (Ohm et al., 2019, line 49; Kudsk et al., 2020 line 71; Mitura et at., 2002, lines 537-538; Sayaslan et al., 2009, line 583; Shevkaniet et al., 2017, line 611. The authors must also clearly state in materials and methods the non-waxy cultivars were used as checks. Another question: in the ANOVA which model did they use the fixed or the random model. They must have used the fixed model and for this their conclusions are valid only for the certain cultivars and years they used. The limiting number of locations (one?) and years (two) excludes the application of the random model, but they must explain it more clearly. Finally, the authors must follow the Journal’s instructions how to write the references in the text (numbers) and in the reference list.
All the above must be consided and I suggest the text to be corrected by a Νatively English Σpeaking editor before submitting the article again.
Author Response
COVER LETTER
Manuscript ID: Plants-1620860
Type of manuscript: Article
Title: Grain Yield Performance and Quality Characteristics of Waxy and Non-Waxy Winter Wheat Cultivars Under High and Low-input Farming Systems
Authors: Jurgita Cesevičienė *, Andrii Gorash, Žilvinas Liatukas, Rita Armonienė, Ruzgas Vytautas, Gražina Statkevičiūtė, Kristina Jaškūnė, Gintaras Brazauskas
Submitted to section: Crop Physiology and Crop Production, https://www.mdpi.com/journal/plants/sections/Crop_Physiology_Crop_Production
Special issue: Impact of Agro-Technological Measures on Quality of Grain https://www.mdpi.com/journal/plants/special_issues/Quality_Grain
Submission date: 16 February 2022
Dear Reviewer and Editor,
We are grateful for valuable advice on how to improve the quality of our manuscript.
We edited the manuscript regarding your recommendations, detailing our changes presented in the tables below. The latest version of the manuscript uploaded to the system has been corrected using “Track Changes”.
Thank you for your time
Regards,
The authors
10 March 2022
Reviewer 2.
Open Review
(x) I would not like to sign my review report
( ) I would like to sign my review report
English language and style
(x) Extensive editing of English language and style required
( ) Moderate English changes required
( ) English language and style are fine/minor spell check required
( ) I don't feel qualified to judge about the English language and style
|
Yes |
Can be improved |
Must be improved |
Not applicable |
|
|
Does the introduction provide sufficient background and include all relevant references? |
( ) |
( ) |
(x) |
( ) |
|
Is the research design appropriate? |
(x) |
( ) |
( ) |
( ) |
|
Are the methods adequately described? |
(x) |
( ) |
( ) |
( ) |
|
Are the results clearly presented? |
( ) |
( ) |
(x) |
( ) |
|
Are the conclusions supported by the results? |
( ) |
( ) |
(x) |
( ) |
Comments and Suggestions for Authors
|
Comments and suggestions for authors |
Appropriate adjustments in manuscript have been made |
|
The article suffers from serious linguistical errors, which must be corrected before any decision. The text is not clear in many cases and has to be written again (i. e. lines 120-122; lines 312-315; 327-330; 343-345; 380-381; 433-435; |
The text was carefully revised and rewritten |
|
Furthermore, there are references in the text which are missing from the reference list (Regina et. Al., 2015, Botticela et. Al., 2018 both in line 41; Graybosch et al., 2003, line 522; Zi et al., 2018, line 525; Bundessortenamt, 2017 in line 560-561. |
Corrected |
|
On the hand, three articles in the reference list ate not referred in the text (lines 730-732; 760-762; 809-812). |
Corrected |
|
Another issue is that the authors do not follow the Journal’s instructions on how to write a references especially in cases that the authors are more than two (Ohm et al., 2019, line 49; Kudsk et al., 2020 line 71; Mitura et at., 2002, lines 537-538; Sayaslan et al., 2009, line 583; Shevkaniet et al., 2017, line 611. |
List of references was corrected according to the instructions of the journal. |
|
The authors must also clearly state in materials and methods the non-waxy cultivars were used as checks. |
Clearly stated in ‘Materials and methods’: non-waxy cultivars were used as control. |
|
Another question: in the ANOVA which model did they use the fixed or the random model. They must have used the fixed model and for this their conclusions are valid only for the certain cultivars and years they used. The limiting number of locations (one?) and years (two) excludes the application of the random model, but they must explain it more clearly. |
A model was specified in more detail: a fixed model was used. |
|
Finally, the authors must follow the Journal’s instructions how to write the references in the text (numbers) and in the reference list. |
Corrected |
|
All the above must be consided and I suggest the text to be corrected by a Νatively English speaking editor before submitting the article again. |
The text has been carefully revised and corrected |
Reviewer 3 Report
The present study is interesting, and valuable for scientific comunity.
Some details need to be slightly modified:
The abstract is not well structured and not clearly presented, according to this research.
Please check the style of text in the whole manuscript.
Lines 138-141: please rephrase, hard to understand
Line 142: estimated?! - shall be calculated
Figure 2 needs to be detailed for each graphic.
Figure 3 needs to be detailed for each graphic.
Figure 4 needs to be detailed for each graphic, and the quality has to be improved.
Figure 5 please re-draft it, legends and axis should be clear and not overapped
The discussion section is clear, understandable and in accordance with the results obtained.
Conclusions section is missing!
The references need to be formated according to journal style.
Author Response
COVER LETTER
Manuscript ID: Plants-1620860
Type of manuscript: Article
Title: Grain Yield Performance and Quality Characteristics of Waxy and Non-Waxy Winter Wheat Cultivars Under High and Low-input Farming Systems
Authors: Jurgita Cesevičienė *, Andrii Gorash, Žilvinas Liatukas, Rita Armonienė, Ruzgas Vytautas, Gražina Statkevičiūtė, Kristina Jaškūnė, Gintaras Brazauskas
Submitted to section: Crop Physiology and Crop Production, https://www.mdpi.com/journal/plants/sections/Crop_Physiology_Crop_Production
Special issue: Impact of Agro-Technological Measures on Quality of Grain https://www.mdpi.com/journal/plants/special_issues/Quality_Grain
Submission date: 16 February 2022
Dear Reviewers and Editor,
We are grateful for valuable advice on how to improve the quality of our manuscript.
We edited the manuscript regarding your recommendations, detailing our changes presented in the tables below. The latest version of the manuscript uploaded to the system has been corrected using “Track Changes”.
Thank you for your time
Regards,
The authors
10 March 2022
Reviewer 3.
Open Review
(x) I would not like to sign my review report
( ) I would like to sign my review report
English language and style
( ) Extensive editing of English language and style required
( ) Moderate English changes required
(x) English language and style are fine/minor spell check required
( ) I don't feel qualified to judge about the English language and style
|
Yes |
Can be improved |
Must be improved |
Not applicable |
|
|
Does the introduction provide sufficient background and include all relevant references? |
(x) |
( ) |
( ) |
( ) |
|
Is the research design appropriate? |
( ) |
(x) |
( ) |
( ) |
|
Are the methods adequately described? |
( ) |
(x) |
( ) |
( ) |
|
Are the results clearly presented? |
( ) |
(x) |
( ) |
( ) |
|
Are the conclusions supported by the results? |
( ) |
( ) |
(x) |
( ) |
Comments and Suggestions for Authors
The present study is interesting, and valuable for scientific comunity.
Some details need to be slightly modified:
|
Comments and suggestions for authors |
Appropriate adjustments in manuscript have been made |
|
The abstract is not well structured and not clearly presented, according to this research. |
The abstract was adjusted to consider the comments of all reviewers. |
|
Please check the style of text in the whole manuscript. |
The text was carefully revised, part of it rewritten |
|
Lines 138-141: please rephrase, hard to understand |
Revised |
|
Line 142: estimated?! - shall be calculated |
Changed to ‘calculated’ |
|
Figure 2 needs to be detailed for each graphic. |
Detailed |
|
Figure 3 needs to be detailed for each graphic. |
Detailed |
|
Figure 4 needs to be detailed for each graphic, and the quality has to be improved. |
Detailed |
|
Figure 5 please re-draft it, legends and axis should be clear and not overapped |
Re-drafted / Detailed |
|
The discussion section is clear, understandable and in accordance with the results obtained. Conclusions section is missing! |
Main conclusions are summed up at the end of the discussion as a separate conclusion section is not mandatory according to the journal instructions. |
|
The references need to be formated according to journal style. |
Corrected |
Round 2
Reviewer 2 Report
No further comments
Author Response
Dear Reviewer,
We are grateful for advice on how to improve the quality of the manuscript.
We appreciate your help. Thank you for your time.
On behalf of the authors
Jurgita Ceseviciene
Reviewer 3 Report
The authors improved the manuscript following reviewers' suggestions/comments. I have no more comments. I agree with the publication of this manuscript.
Author Response

(The authors gave the same response as above.)
